# Redox-dependent rearrangements of the NiFeS cluster of carbon monoxide dehydrogenase

Elizabeth C Wittenborn[1†], Mériem Merrouch[2], Chie Ueda[3], Laura Fradale[2], Christophe Léger[2], Vincent Fourmond[2], Maria-Eirini Pandelia[3], Sébastien Dementin[2*], Catherine L Drennan[1,4,5,6*]

[1]Department of Chemistry, Massachusetts Institute of Technology, Cambridge, United States; [2]Aix Marseille Univ, CNRS, Laboratoire de Bioénergétique et Ingénierie des Protéines, Marseille, France; [3]Department of Biochemistry, Brandeis University, Waltham, United States; [4]Department of Biology, Massachusetts Institute of Technology, Cambridge, United States; [5]Howard Hughes Medical Institute, Massachusetts Institute of Technology, Cambridge, United States; [6]Bio-inspired Solar Energy Program, Canadian Institute for Advanced Research, Toronto, Canada

*For correspondence:
dementin@imm.cnrs.fr (SD);
cdrennan@mit.edu (CLD)

Present address: [†]California Institute for Quantitative Biosciences, University of California, Berkeley, United States

Competing interests: The authors declare that no competing interests exist.

**Abstract** The C-cluster of the enzyme carbon monoxide dehydrogenase (CODH) is a structurally distinctive Ni-Fe-S cluster employed to catalyze the reduction of $CO_2$ to CO as part of the Wood-Ljungdahl carbon fixation pathway. Using X-ray crystallography, we have observed unprecedented conformational dynamics in the C-cluster of the CODH from *Desulfovibrio vulgaris*, providing the first view of an oxidized state of the cluster. Combined with supporting spectroscopic data, our structures reveal that this novel, oxidized cluster arrangement plays a role in avoiding irreversible oxidative degradation at the C-cluster. Furthermore, mutagenesis of a conserved cysteine residue that binds the C-cluster in the oxidized state but not in the reduced state suggests that the oxidized conformation could be important for proper cluster assembly, in particular Ni incorporation. Together, these results lay a foundation for future investigations of C-cluster activation and assembly, and contribute to an emerging paradigm of metallocluster plasticity.
DOI: https://doi.org/10.7554/eLife.39451.001

## Introduction

Roughly half of all enzymes make use of metal centers to expand their chemical repertoire (*Waldron et al., 2009*). Among the most fascinating of the metallocofactors used for such purposes are Fe-S clusters, which are thought to be the most ancient biological cofactors and which enable chemical transformations ranging from simple electron transfer events to the formation and cleavage of carbon-carbon bonds (*Rees and Howard, 2003*; *Beinert et al., 1997*). Complex Fe-S clusters, containing alternative metal ions and/or expanded metal frameworks, such as the FeMo-cofactor of nitrogenase, the H-cluster of Fe-Fe hydrogenase, and the C-cluster of Ni-dependent carbon monoxide dehydrogenase (CODH), catalyze fundamental redox conversions that are thought to have enabled early life on Earth (*Rees and Howard, 2003*; *Rees, 2002*). Given their structural complexity and the essential nature of the reactions they catalyze, these clusters have collectively been termed the 'great clusters' of biology (*Rees, 2002*).

The great clusters, and the proteins that house them, have become the focus of extensive mechanistic and structural investigation in the hopes of yielding new applications in clean energy production and bioremediation. In particular, CODH catalyzes the interconversion of the gaseous pollutant

**eLife digest** Life relies on countless chemical reactions, almost all of which need to be sped up by enzymes. About half of all enzymes carry metal ions that expand the range of the reactions that they can catalyze. In some enzymes these metal ions assemble with sulfur ions to form so-called metalloclusters. These structures can carry out many different types of reactions, including converting simple forms of elements like nitrogen and carbon into other forms that can be used to make more complicated biological molecules.

One enzyme that contains metalloclusters is carbon monoxide dehydrogenase. Known as CODH for short, this enzyme uses a metallocluster called the "C-cluster" to interconvert two gases: the pollutant carbon monoxide and the greenhouse gas carbon dioxide. CODH enzymes are found inside certain bacteria, but they are also of interest for humans, who wish to use them to remove the harmful gases from the environment. But this is not as simple as it may at first seem: CODH enzymes usually become inactive when exposed to air because the metalloclusters fall apart in the presence of oxygen. One CODH enzyme from a widespread bacterium called *Desulfovibrio vulgaris*, however, is an attractive target for industrial use because it can tolerate oxygen better. Yet, it is still unclear why this enzyme does not get inactivated the way other CODHs do.

Wittenborn et al. have now characterized the CODH enzyme from *D. vulgaris* in more depth via a technique called X-ray crystallography, which can reveal the location of individual atoms within a molecule. By a happy accident, the structures revealed that the C-cluster can adopt a dramatically different arrangement of metal and sulfur ions after being exposed to oxygen. This rearrangement is fully reversible; when oxygen is removed, the metal and sulfur ions move back to their normal positions. This ability to flip between different arrangements appears to protect the metallocluster from losing its metal ions when exposed to oxygen.

By providing structural snapshots of how CODH responds to oxygen these results provide a more complete understanding of an enzyme that plays a key role in the global carbon cycle. This understanding could help scientists to develop bioremediation tools to remove carbon monoxide and carbon dioxide from the atmosphere and to engineer bacteria to capture carbon to make biofuels.

DOI: https://doi.org/10.7554/eLife.39451.002

CO and the greenhouse gas $CO_2$, leading to the removal of an estimated $10^8$ tons of CO from the lower atmosphere each year and making it an attractive remediation tool (*Bartholomew and Alexander, 1979*). The anaerobic, Ni-dependent CODH has a homodimeric structure containing a total of five metalloclusters, called the B-, C-, and D-clusters. The C-cluster is the site of $CO/CO_2$ interconversion and is composed of a [Ni-3Fe-4S] cubane connected through a linking sulfide ($S_L$) to a unique iron site ($Fe_u$) (*Figure 1*) (*Drennan et al., 2001*; *Dobbek et al., 2001*). Comprehensive spectroscopic analyses have revealed the basic redox states and kinetic properties of this complex metallocluster, and crystal structures with substrates and inhibitors bound have provided snapshots along the reaction pathway (*Figure 2*) (*Jeoung and Dobbek, 2007*; *Gong et al., 2008*; *Kung et al., 2009*; *Jeoung and Dobbek, 2009*; *Fesseler et al., 2015*; *Lindahl et al., 1990*; *Kumar et al., 1993*; *Anderson and Lindahl, 1994*; *Anderson and Lindahl, 1996*; *Seravalli et al., 1997*; *Fraser and Lindahl, 1999*; *Chen et al., 2003*; *Seravalli and Ragsdale, 2008*; *Drennan and Peters, 2003*). These studies have revealed the C-cluster to take on four discrete redox states termed $C_{ox}$, $C_{red1}$, $C_{int}$, and $C_{red2}$ (*Lindahl et al., 1990*; *Kumar et al., 1993*; *Anderson and Lindahl, 1994*; *Anderson and Lindahl, 1996*; *Seravalli et al., 1997*; *Fraser and Lindahl, 1999*). The most widely accepted mechanism of CO oxidation involves a one-electron reductive activation of the inactive $C_{ox}$ state to $C_{red1}$ followed by a catalytic cycle involving conversion between $C_{red1}$ and $C_{red2}$ (*Figure 2*) (*Lindahl et al., 1990*; *Kim et al., 2004*; *Lindahl, 2008*). Despite this relatively unified understanding of CO oxidation activity, there are still many gaps in our understanding of this complicated enzyme that are limiting with regard to both our understanding of CODH biochemistry and potential applications of CODH in industrial settings. In particular, there has been a push to characterize enigmatic redox states and also to probe the effects of molecular oxygen on enzyme activity (*Merrouch et al., 2015*; *Wang et al., 2015*; *Domnik et al., 2017*). Here, we report the crystal structure of the CODH from

*Desulfovibrio vulgaris* (*Dv*CODH) (*Hadj-Saïd et al., 2015*), which reveals a surprising and unprecedented conformational rearrangement of metal ions in the C-cluster and provides the first visualization of the cluster in an oxidized state. Through combined structural and spectroscopic data, we show that conversion between the oxidized and reduced states of the cluster is reversible, consistent with previous electrochemical investigations (*Merrouch et al., 2015*). We further consider the implications of these findings in terms of oxygen sensitivity and cluster assembly and with respect to the other great clusters in biology.

## Results and discussion

The overall fold and cluster placement of *Dv*CODH is highly similar to other structurally characterized CODHs (*Figure 1—figure supplement 1*) (*Drennan et al., 2001; Dobbek et al., 2001; Domnik et al., 2017; Doukov et al., 2002; Darnault et al., 2003*). One noteworthy difference with respect to other CODHs is the identity of the D-cluster, a solvent-exposed Fe-S cluster at the dimer interface that serves as an electron conduit to the surface of the protein. Instead of the expected [4Fe-4S] cluster, (*Drennan et al., 2001; Dobbek et al., 2001; Domnik et al., 2017; Doukov et al., 2002; Darnault et al., 2003*) the electron density is consistent with a [2Fe-2S] cluster (*Figure 1—figure supplement 2*) as is the placement of cysteine residues in the primary structure. CODH sequence alignments reveal that instead of a C-$X_7$-C D-cluster binding motif, *Dv*CODH, as well as several uncharacterized CODHs, have a C-$X_2$-C motif (*Figure 1—figure supplement 2*). This shortened C-$X_2$-C motif appears to constrain the geometry of the ligating cysteine residues such that coordination to a [4Fe-4S] cluster is not possible. Instead, the cysteine positions are ideally suited for coordination of a [2Fe-2S] cluster.

In the present work, we determined two structures of as-isolated *Dv*CODH using two independent protein batches and much to our surprise, we observe different conformations of the C-cluster in each structure. A 2.50 Å resolution structure of as-isolated *Dv*CODH (determined using protein batch 1) displays the canonical [Ni-3Fe-4S]-$Fe_u$ C-cluster (*Figure 3a; Figure 3—figure supplement*

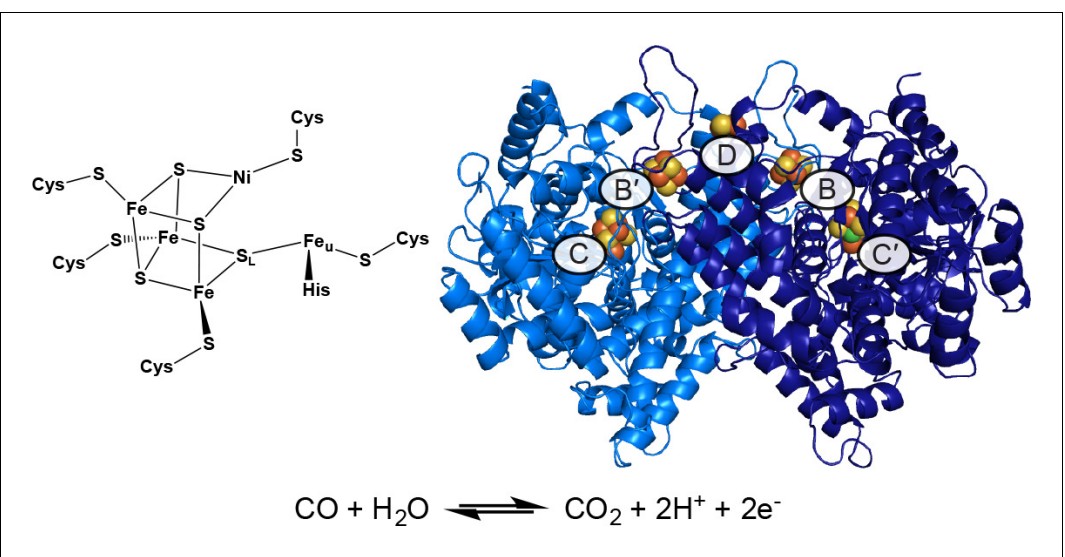

$$CO + H_2O \rightleftharpoons CO_2 + 2H^+ + 2e^-$$

**Figure 1.** Structure of the C-cluster and CODH. Protein structure shown in ribbon representation in blue with metalloclusters labeled and shown as spheres; Ni in green, Fe in orange, S in yellow. Reaction catalyzed by CODH is shown below.

DOI: https://doi.org/10.7554/eLife.39451.003

The following figure supplements are available for figure 1:

**Figure supplement 1.** Overall structure of *Dv*CODH.

DOI: https://doi.org/10.7554/eLife.39451.004

**Figure supplement 2.** *Dv*CODH contains a distinguishing [2Fe-2S] D-cluster.

DOI: https://doi.org/10.7554/eLife.39451.005

**Figure 2.** Proposed mechanism of CO oxidation at the CODH C-cluster. The catalytic cycle begins with the C-cluster in the $C_{red1}$ redox state (a one-electron reduced state of the C-cluster) with $H_2O$ bound to $Fe_u$ (*Jeoung and Dobbek, 2007*; *Kung et al., 2009*). CO binds to the cluster in a bent binding mode (*Gong et al., 2008*; *Kung et al., 2009*) and then undergoes a 'carbon shift,' positioning the carbonyl carbon atom for nucleophilic attack by a $Fe_u$-bound hydroxide, formed by loss of a proton from $Fe_u$-bound $H_2O$ to a catalytic base, which is proposed to be a conserved lysine residue (*Drennan et al., 2001*; *Dobbek et al., 2001*; *Kim et al., 2004*). The resulting COOH-type species is deprotonated by a second catalytic base, proposed to be an active site histidine residue, to form a metallocarboxylate species (*Drennan et al., 2001*; *Dobbek et al., 2001*; *Jeoung and Dobbek, 2009*; *Fesseler et al., 2015*; *Chen et al., 2003*; *Kim et al., 2004*). CO oxidation reduces the C-cluster by two electrons forming the $C_{red2}$ state (a species that is two-electrons more reduced than $C_{red1}$) and $CO_2$ is released (*Jeoung and Dobbek, 2007*; *Kumar et al., 1993*; *Anderson and Lindahl, 1996*). We note that, although $C_{red2}$ formation and $CO_2$ release have been drawn concomitantly, the rate of $CO_2$ release has been shown to be slower than the rate of cluster reduction (*Seravalli and Ragsdale, 2008*). For the next round of turnover, the cluster undergoes a two-electron oxidation, reforming the $C_{red1}$ state. Cluster oxidation is thought to proceed through two single-electron transfer events via an intermediate $C_{int}$ redox state as electrons flow through the B- and D-clusters to an external redox partner, such as ferredoxin. The $C_{ox}$ redox state is one electron more oxidized than $C_{red1}$. States of the C-cluster that have been visualized crystallographically are indicated with a red asterisk.
DOI: https://doi.org/10.7554/eLife.39451.006

1a; *Supplementary file 1*) (*Drennan et al., 2001*; *Gong et al., 2008*; *Doukov et al., 2002*; *Darnault et al., 2003*); the [Ni-3Fe-5S]-$Fe_u$ state that was observed in structures of *Carboxydothermus hydrogenoformans* CODH-II (*Dobbek et al., 2001*; *Dobbek et al., 2004*) is no longer thought to be catalytically relevant (*Jeoung and Dobbek, 2007*; *Kung et al., 2009*; *Drennan and Peters, 2003*; *Feng and Lindahl, 2004*). The ligation of the C-cluster is conserved in *Dv*CODH (*Drennan et al., 2001*; *Dobbek et al., 2001*; *Gong et al., 2008*; *Domnik et al., 2017*; *Doukov et al., 2002*; *Darnault et al., 2003*) with four cysteines ligating the cubane portion of the cluster (Cys519 is the Ni ligand); one histidine (His266) and one cysteine (Cys302) ligate $Fe_u$ (*Figure 3a*; *Figure 3—figure supplement 1a*).

The second, higher-resolution (1.72 Å) structure of as-isolated *Dv*CODH (determined using protein batch 2) reveals a novel arrangement of ions within the C-cluster that was confirmed using anomalous diffraction data (*Figure 3b*; *Supplementary file 1*). In this structure, Ni, $Fe_u$, and $S_L$ are shifted, accompanied by conformational changes of several amino acid side chains, while the positions of the remaining three Fe and three S ions are unchanged. The Ni ion is bound in the site formerly occupied by $Fe_u$, coordinated by His266 and Cys302 (*Figure 3b*; *Figure 3—figure supplement 1b*). The Ni is additionally ligated by Cys519 and Lys556 (*Figure 3b*; *Figure 3—figure supplement 1b*). As mentioned above, Cys519 serves as a ligand to Ni in the canonical C-cluster and here adopts an alternative rotamer conformation such that coordination to Ni is maintained (*Figure 3b*; *Figure 3—figure supplement 1b*). The occupancy of the alternative Cys519 conformation correlates with the occupancy of Ni, and both have been refined at an atomic occupancy of

70%, in general agreement with the metal analysis result of 0.5 Ni per monomer for the sample that was crystallized (*Supplementary file 2*). Lys556 is highly conserved and does not normally coordinate to the C-cluster, but is instead the proposed general base catalyst for deprotonation of water during CO oxidation (*Figure 2*) (*Drennan et al., 2001*; *Dobbek et al., 2001*; *Kim et al., 2004*). Here, the lysine amine group comes within 2.5 Å of Ni. Together, His266, Cys302, Cys519, and Lys556 ligate the Ni in this altered cluster in a highly distorted tetrahedral coordination geometry that is reminiscent of the geometry of the Ni site in Ni-Fe hydrogenases (*Volbeda et al., 1995*).

Concomitant with the shift of the Ni ion, $Fe_u$ and $S_L$ also undergo changes in their coordination environments while remaining associated with the [3Fe-3S] partial cubane (*Figure 3b*; *Figure 3—figure supplement 1b*). In addition to interaction with $S_L$, $Fe_u$ is coordinated by Cys302, which forms a bridging interaction with Ni, and by a conserved cysteine residue (Cys301) that does not normally serve as a ligand to the C-cluster, resulting in an apparent three-coordinate geometry around $Fe_u$ (*Figure 3b*; *Figure 3—figure supplement 1b*). The shift in the positions of $Fe_u$ and $S_L$ results in the

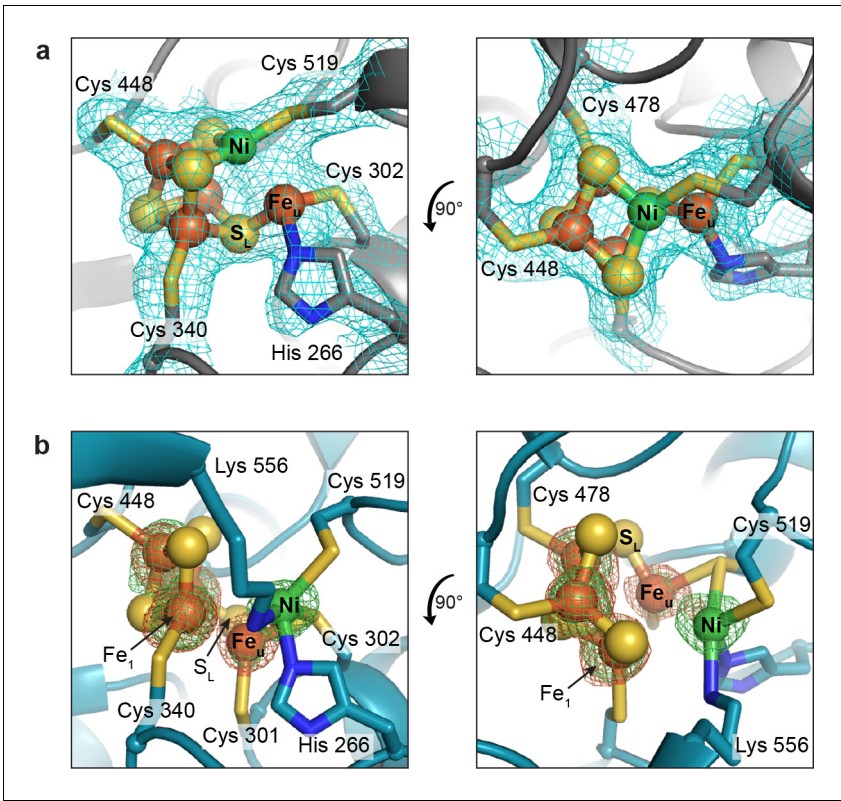

**Figure 3.** The *Dv*CODH C-cluster adopts alternative conformations. (a) C-cluster with simulated annealing composite omit electron density map (contoured to 1σ) as observed in the 2.50 Å resolution structure of as-isolated *Dv*CODH (batch 1). The cluster adopts the canonical conformation, ligated by the indicated amino acid residues. Lys556 does not ligate to the cluster in this conformation and has been omitted here for clarity. The positioning of Lys556 is shown in *Figure 3—figure supplement 1a*. (b) C-cluster as observed in the 1.72 Å resolution structure of as-isolated *Dv*CODH (batch 2). Ni, $Fe_u$, and $S_L$ have shifted relative to the canonical cluster and Cys301 and Lys556 form new interactions to the cluster. Fe and Ni anomalous difference maps, calculated from data collected at Fe and Ni peak wavelengths (7130 and 8360 eV, respectively), are shown as green and orange mesh contoured to 6σ and 5σ, respectively. Protein is shown in ribbon representation with ligating amino acid residues as sticks and C-cluster as spheres and sticks; Ni in green, Fe in orange, S in yellow, N in blue.
DOI: https://doi.org/10.7554/eLife.39451.007

The following figure supplements are available for figure 3:

**Figure supplement 1.** Views of the alternative C-cluster arrangements.
DOI: https://doi.org/10.7554/eLife.39451.008

**Figure supplement 2.** A peak of residual electron density is present within the alternative C-cluster.
DOI: https://doi.org/10.7554/eLife.39451.009

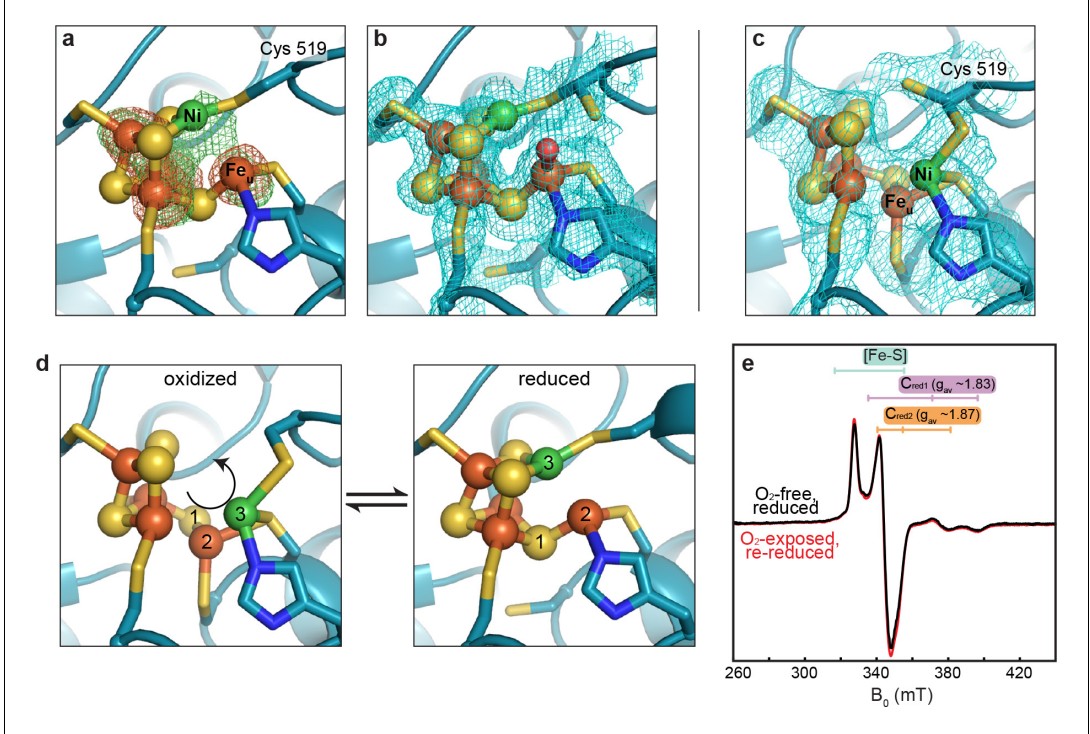

**Figure 4.** The *Dv*CODH C-cluster undergoes reversible, redox-dependent conformational changes. (**a**) C-cluster of dithionite-soaked *Dv*CODH crystal (batch 2). Ni and Fe anomalous difference maps are shown as green and orange mesh, respectively, and contoured to 5σ. (**b**) C-cluster of dithionite-soaked *Dv*CODH crystal (batch 2) with simulated annealing composite omit electron density map contoured to 1σ. A water molecule (red sphere) is observed bound to $Fe_u$. (**c**) C-cluster of reduced and then air-exposed *Dv*CODH crystal (batch 2) with simulated annealing composite omit electron density map contoured to 1σ. Lys556 has not been shown for simplicity. (**d**) Illustration of the conversion between the oxidized and reduced states of the C-cluster. Arrow and numbers indicate the assumed direction of metal ion movement from the oxidized state to the reduced state. (**e**) Continuous-wave X-Band EPR spectra of dithionite reduced *Dv*CODH before (black trace) and after (red trace) exposure to air. Experimental conditions: mw power = 0.2 mW, mw frequency = 9.34 GHz, modulation amplitude 1 mT, temperature = 10 K. For panels a-d, protein is shown in ribbon representation with ligating amino acid residues as sticks and C-cluster as spheres and sticks; Ni in green, Fe in orange, S in yellow, N in blue, O in red.
DOI: https://doi.org/10.7554/eLife.39451.010

The following video and figure supplement are available for figure 4:

**Figure supplement 1.** The D- and B-clusters of *Dv*CODH are resistant to oxidative damage.
DOI: https://doi.org/10.7554/eLife.39451.011

**Figure 4—video 1.** The C-cluster of *Dv*CODH undergoes redox-dependent conformational changes.
DOI: https://doi.org/10.7554/eLife.39451.012

loss of an interaction between $S_L$ and a second Fe ion of the cubane ($Fe_1$), leaving $Fe_1$ with three coordinating ligands, Cys340 and two cubane sulfides. A small peak of residual electron density at a site bridging $Fe_u$ and $Fe_1$ is present in one of the protein chains in the asymmetric unit (*Figure 3—figure supplement 2*). The occupancy of this site is low (<30%) precluding identification of the atom/ion. However, if fully occupied, the atom/ion would complete the tetrahedral geometry around these Fe atoms. In addition to changes in Fe coordination, the altered conformation of the C-cluster also involves changes in sulfide coordination state. In particular, the two cubane S ions that coordinate Ni in the canonical cluster are left in a possibly unstable state in which they could be susceptible to protonation or loss as free sulfide ions (*Crack et al., 2006*). In our structure, however, we see no evidence of degradation at these sites, likely due to inaccessibility to solvent or other protective features of the protein environment, in analogy to the case of stable [3Fe-4S] clusters.

We next investigated whether this altered C-cluster state is redox dependent. Pre-formed crystals of as-isolated *Dv*CODH (batch 2, containing the altered cluster) were incubated with the reductant sodium dithionite. Strikingly, the resulting crystal structure displays the canonical C-cluster with Ni, $Fe_u$, and $S_L$ rearranged into their catalytically-relevant positions (*Figure 4a,b; Supplementary file 1*).

To examine whether this metal rearrangement is reversible upon oxidation, crystals of reduced *Dv*CODH were taken from the anaerobic chamber and incubated under ambient atmospheric conditions. Remarkably, oxidation of the reduced C-cluster by exposure to $O_2$ results in reformation of the unusual cluster architecture (*Figure 4c*; *Supplementary file 1*), whereas both the D- and B-clusters remain intact (*Figure 4—figure supplement 1*). Together, these results suggest that this altered cluster is an oxidized form of the C-cluster and that this multi-metal ion rearrangement is reversible (*Figure 4d*, *Figure 4—video 1*). Most likely, a fortuitous oxidation event, affecting *Dv*CODH batch 2, initially allowed us to obtain the first visualization of an oxidized state of the C-cluster.

Given the apparent ability of *Dv*CODH to undergo fully reversible oxidation/reduction events *in crystallo*, we used electron paramagnetic resonance (EPR) spectroscopy to determine the effect of oxidation on the enzyme in solution. First, an EPR spectrum was recorded on a sample of dithionite-reduced *Dv*CODH. This spectrum exhibits resonances characteristic of the one-electron reduced B- and D-clusters (centered around g ~ 2) and of the $C_{red1}$ and $C_{red2}$ forms of the C-cluster ($g_{av}$ ~ 1.83 and 1.87, respectively; see *Figure 2*), as has been previously observed for *Dv*CODH (*Hadj-Saïd et al., 2015*) (*Figure 4e*, black trace). A parallel dithionite-reduced sample was incubated under ambient atmospheric conditions to mimic treatment of the *Dv*CODH crystals (EPR-silent, data not shown). The oxygen-exposed sample was then re-reduced with dithionite and the EPR spectrum was recorded, revealing full recovery of the previously observed signals (*Figure 4e*, red trace). Combined, our crystal structures and EPR data reveal that the metalloclusters of *Dv*CODH are not degraded upon oxidation and that the C-cluster avoids degradation by adopting an alternative, stable, oxidized conformation. Consistent with these results, *Dv*CODH was recently shown to regain activity upon chemical or electrochemical reduction following exposure to molecular oxygen (*Merrouch et al., 2015*). In this respect, the rearranged C-cluster scaffold can be thought of as a 'safety net' for retaining cluster ions upon oxygen exposure, and may explain, at least in part, the ability of certain CODHs to recover activity following oxidation in air (*Merrouch et al., 2015*; *Wang et al., 2015*; *Domnik et al., 2017*). This kind of safety net could improve the ability of an organism to rapidly recover CODH activity after transient exposure to oxic conditions.

One of the more noteworthy aspects of the oxidized C-cluster is that cluster ligation involves one residue (Cys301) that is strictly conserved in CODHs but is not a ligand to the canonical C-cluster. Interestingly, previous work on the heterotetrameric CODH/acetyl-CoA synthase from *Moorella thermoacetica* (*Mt*CODH/ACS) showed that mutation of the equivalent cysteine residue (Cys316) to serine resulted in an inactive CODH that appeared to lack an intact C-cluster (*Kim et al., 2004*).

To probe the effect of Cys301 in *Dv*CODH, a *Dv*CODH(C301S) variant was produced and characterized. Similar to what was observed with *Mt*CODH/ACS (*Kim et al., 2004*), *Dv*CODH (C301S) is inactive and does not contain Ni, as assessed by CO oxidation activity assays and inductively coupled plasma optical emission spectroscopy (ICP-OES), respectively (*Supplementary file 2*). Unlike wild-type *Dv*CODH, *Dv*CODH(C301S) cannot be activated by Ni under reducing conditions, analogous to what has been observed previously for *Dv*CODH samples grown in the absence of the C-cluster maturation factor CooC (*Hadj-Saïd et al., 2015*; *Merrouch et al., 2018*).

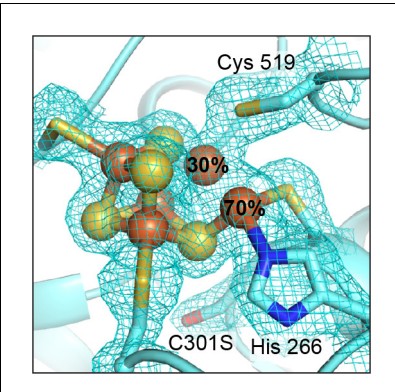

**Figure 5.** The *Dv*CODH(C301S) C-cluster is a mixture of species. The C-cluster has been refined with an alternative conformation of $Fe_u$. At 70% occupancy, $Fe_u$ is ligated by His266 and Cys302 in its canonical binding site. At 30% occupancy, $Fe_u$ is incorporated into the Fe-S cubane portion of the cluster. $2F_O - F_C$ electron density contoured to 1σ. Protein is shown in ribbon representation with ligating amino acid residues as sticks and C-cluster as spheres and sticks; Fe in orange, S in yellow, N in blue, O in red.

DOI: https://doi.org/10.7554/eLife.39451.013

The following figure supplement is available for figure 5:

**Figure supplement 1.** The *Dv*CODH(C301S) C-cluster is a mixture of species.

DOI: https://doi.org/10.7554/eLife.39451.014

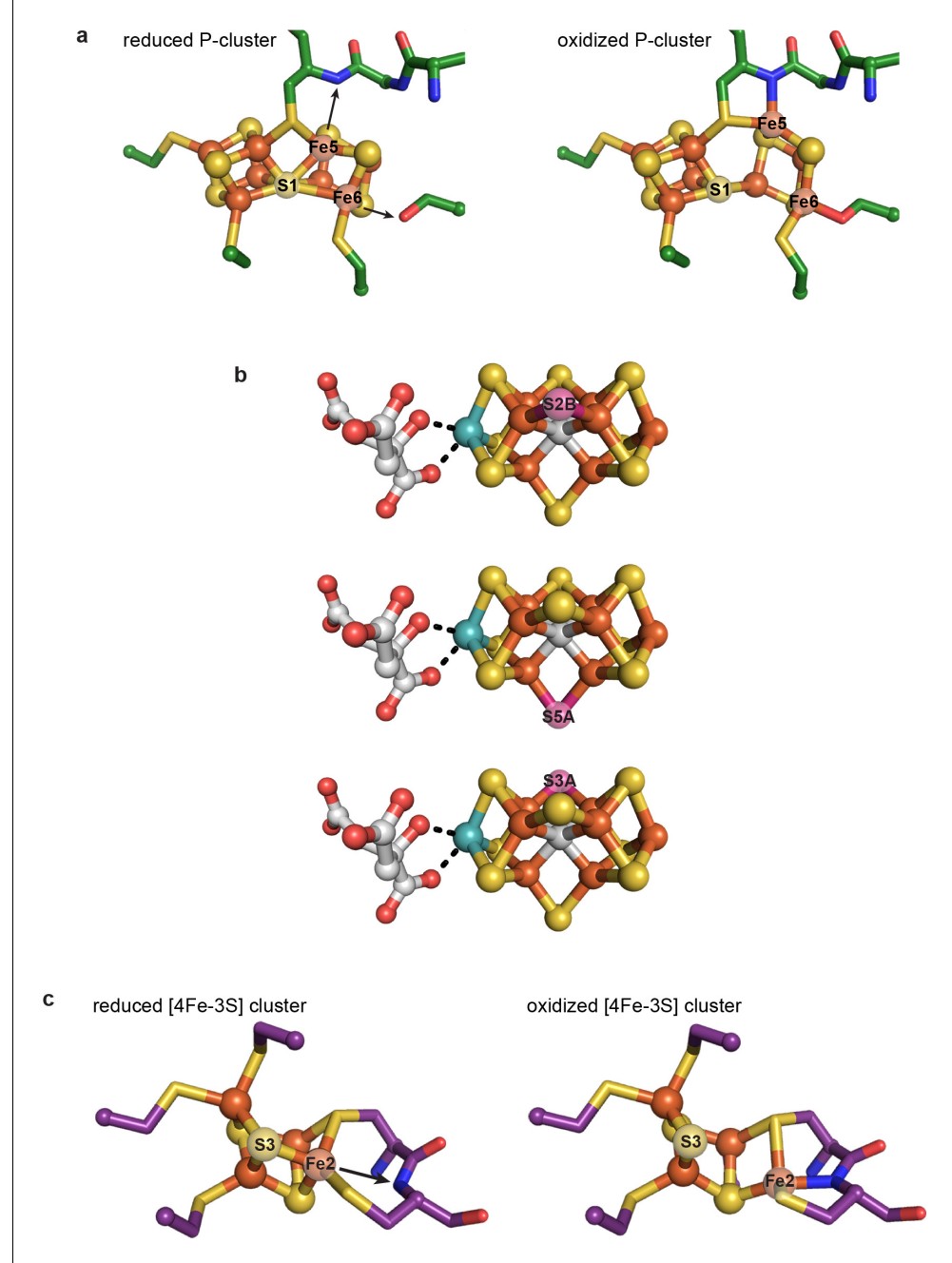

**Figure 6.** Previously characterized structural lability in Fe-S clusters. (**a**) The reduced and oxidized P-cluster of nitrogenase. Upon oxidation, Fe5 and Fe6 lose coordination to the central S1 ion, moving distances of 1.4 and 0.9 Å, respectively. Arrows in the left hand panel show the direction of Fe ion movement. Protein is shown as sticks with P-cluster as spheres and sticks; Fe in orange, S in yellow, C in green, N in blue, O in red. PDB IDs: 3MIN (reduced) and 2MIN (oxidized). (**b**) The FeMo-cofactor of nitrogenase undergoes turnover-dependent rearrangements, observed as the movement of an artificially-incorporated Se atom. Under turnover conditions, Se migrates through the cluster from position S2B to S5A to S3A. FeMo-cofactor is shown as spheres and sticks; Fe in orange, S in yellow, Se in pink, Mo in teal, C in grey, O in red. PDB ID: 5BVG. (**c**) A unique [4Fe-3S] cluster is present in the $O_2$-tolerant membrane-bound hydrogenase. Upon oxidation, Fe2 loses coordination to S3 and becomes coordinated by a backbone amide group of the protein. Arrow in the left hand panel shows the direction of Fe ion movement. Protein is shown as sticks with [4Fe-3S] cluster as spheres and sticks; Fe in orange, S in yellow, C in purple, N in blue, O in red. PDB IDs: 3AYX (reduced), 2AYY (oxidized).
DOI: https://doi.org/10.7554/eLife.39451.015

To investigate the architecture of a non-activatable C-cluster, we determined the crystal structure of *Dv*CODH(C301S) to 2.0 Å resolution (*Supplementary file 1*) and discovered an intact [3Fe-4S] C-cluster core with an $Fe_u$ ion that adopts alternative conformations (*Figure 5*). Approximately 70% of $Fe_u$ is in its canonical position coordinated by His266 and Cys302, whereas approximately 30% is in the Ni-cubane site (see Methods; *Figure 5—figure supplement 1*). Thus, in the absence of the non-canonical $Fe_u$-ligand Cys301, the cluster cannot be activated by Ni and $Fe_u$ appears to be free to occupy multiple sites. Taken together, these data suggest that the novel structure of the C-cluster that we observe here, with $Fe_u$ coordinated by Cys301, is relevant to processes beyond oxidation, specifically Ni incorporation.

Regardless of its role(s), the dramatic rearrangement observed for the C-cluster adds to a growing appreciation that the ions of great clusters are in fact mobile. The oxidized P-cluster of nitrogenase adopts an open conformation relative to its reduced form through the outward movement of two Fe ions by 1.4 and 0.9 Å with accompanying loss of S coordination in a rearrangement that is proposed to couple proton transfer to electron transfer (*Figure 6a*) (*Peters et al., 1997*). More recently, Rees and coworkers demonstrated that both the inhibitor CO and an artificially-incorporated Se atom are able to displace a S atom of the cluster and furthermore that, under turnover conditions, the Se atom migrates around the cluster, sampling S positions (*Figure 6b*) (*Spatzal et al., 2014*; *Spatzal et al., 2015*). Very recently, it was also revealed that the same S atom of the VFe-cofactor (containing V in place of Mo) is displaced by a NH ligand, suggesting that cluster dynamics are likely key in catalysis (*Sippel et al., 2018*). Additionally, an $O_2$-tolerant membrane-bound hydrogenase was shown to have a novel [4Fe-3S] cluster that undergoes redox-dependent structural changes as part of an $O_2$-tolerance mechanism (*Fritsch et al., 2011*; *Shomura et al., 2011*); one Fe ion moves ~1.6 Å upon cluster oxidation and becomes coordinated by a protein backbone amide group (*Figure 6c*) (*Shomura et al., 2011*). In the present work, the metal migration of C-cluster atoms is more dramatic than in these other examples, with Ni moving ~3 Å and adopting an entirely new coordination environment, and $Fe_u$ and $S_L$ moving ~1.9 and 2.6 Å, respectively, with $Fe_u$ also taking on a new coordination environment. This C-cluster rearrangement from oxidized to reduced appears to involve what amounts to a 'molecular cartwheel' with Ni, $Fe_u$, and $S_L$ following the same trajectory to end up in their canonical positions (*Figure 4d*, *Figure 4—video 1*).

In summary, X-ray crystallography has provided views of the 'great clusters' of biology, allowing us to marvel at these incredible metallic frameworks that capture and make use of CO, $H_2$, and $N_2$ gases. We are increasingly finding that these frameworks should not be thought of as rigid scaffolds, but rather as labile assemblies of metal with sulfide. The full nature and significance of this metallocluster lability is just now beginning to emerge and the roles appear to be diverse, including catalysis, electron transfer, protection from oxygen damage, and possibly cluster assembly. The one consistency is that these great clusters continue to surprise us.

## Materials and methods

### Key resources table

| Reagent type (species) or resource | Designation | Source or reference | Identifiers | Additional information |
|---|---|---|---|---|
| Gene (*Desulfovibrio vulgaris* str. Hildenborough) | *cooS* | NA | NCBI:2795474 | |
| Gene (*Desulfovibrio vulgaris* str. Hildenborough) | *cooC* | NA | NCBI:2795475 | |
| Cell line (*Desulfovibrio fructosovorans* str. MR400) | *Desulfovibrio fructosovorans* str. MR400 | PMID:1943706 | | |
| Recombinant DNA reagent | modified pBGF4 shuttle vector | PMID:26255854 | | |

*Continued on next page*

*Continued*

| Reagent type (species) or resource | Designation | Source or reference | Identifiers | Additional information |
|---|---|---|---|---|
| Sequence-based reagent | C301S forward primer | Eurogentec | | ACATCAACGTGGCGGGG CTATCCTGCACGGGTA ACGAACTGCTC |
| Sequence-based reagent | C301S reverse primer | Eurogentec | | GAGCAGTTCGTTACCCG TGCAGGATAGCCCCGCC ACGTTGATGT |
| Peptide, recombinant protein | *Dv*CODH | PMID:26255854 | | |
| Peptide, recombinant protein | *Dv*CODH(C301S) | this paper | | *Dv*CODH variant produced in the lab of Dr. C. Léger as described in Methods |
| Software, algorithm | XDS/XSCALE | PMID:20124692 | RRID:SCR_015652 | |
| Software, algorithm | Phaser | PMID:19461840 | RRID:SCR_014219 | |
| Software, algorithm | Scupltor | PMID:21460448 | | |
| Software, algorithm | Schwarzenbacher algorithm | PMID:15213384 | | |
| Software, algorithm | Coot | PMID:20383002 | RRID:SCR_014222 | |
| Software, algorithm | Phenix | PMID:20124702 | RRID:SCR_014224 | |
| Software, algorithm | TLS parameterization | PMID:16552146 | | |
| Software, algorithm | MolProbity | PMID:20057044 | RRID:SCR_014226 | |
| Software, algorithm | PyMOL | www.pymol.org/ | RRID:SCR_000305 | |
| Software, algorithm | EasySpin | PMID:16188474 | | |

## Protein preparation, metal analysis, and activity assays of DvCODH and DvCODH(C301S)

*Dv*CODH was expressed in the presence of the C-cluster maturation factor CooC, as described previously (*Hadj-Saïd et al., 2015*). Briefly, the *D. vulgaris* genes encoding CODH (*cooS*) and the CooC maturase (*cooC*) were cloned into modified pBGF4 shuttle vectors under the control of the promoter of the *Desulfovibrio fructosovorans* Ni-Fe hydrogenase operon. The CODH construct was N-terminally strep-tagged. A construct encoding *Dv*CODH(C301S) was generated from the wild-type sequence by site-directed mutagenesis (forward primer ACATCAACGTGGCGGGGCTA TCCTGCACGGGTAACGAACTGCTC, reverse primer GAGCAGTTCGTTACCCGTGCAGGA-TAGCCCCGCCACGTTGATGT; mutation underlined). Protein was expressed in *D. fructosovorans* str. MR400 (*Rousset et al., 1991*) and purified under anaerobic conditions in a Jacomex anaerobic chamber (100% $N_2$ atmosphere) by affinity chromatography on Strep-Tactin Superflow resin, as described previously (*Hadj-Saïd et al., 2015*). Protein concentrations were determined by amino acid analysis at the Centre for Integrated Structural Biology (Grenoble, France). Metal content (*Supplementary file 2*) was analyzed by inductively coupled plasma optical emission spectroscopy (ICP-OES). CO oxidation activity was assayed at 37 °C by monitoring the reduction of methyl viologen at 604 nm ($\varepsilon$ = 13.6 mM$^{-1}$·cm$^{-1}$), as described previously (*Hadj-Saïd et al., 2015*) (*Supplementary file 2*).

## Crystallization of DvCODH and DvCODH(C301S)

All crystals were grown using as-isolated protein samples (i.e., samples were not activated with $NiCl_2$ and sodium dithionite prior to crystallization). Crystals were grown anaerobically in an $N_2$ atmosphere at 21 °C by hanging drop vapor diffusion in an MBraun anaerobic chamber. Crystals belonging to space group $P2_12_12_1$ were obtained as follows: A 1 µL aliquot of as-isolated protein (10 mg/mL in 100 mM Tris-HCl pH 8) was combined with 1 µL of precipitant solution (1.0–1.1 M ammonium tartrate dibasic pH 7, 6–9% (v/v) glycerol) on a glass cover slide and sealed over a reservoir containing 500 µL of precipitant solution. Diffraction quality crystals grew in 2–10 d. Crystals were soaked in a cryo-protectant solution containing 1.0–1.2 M ammonium tartrate dibasic pH 7, 25% (v/v) glycerol and cryo-cooled in liquid nitrogen.

Crystals belonging to either space group $P2_1$ or $P1$ were obtained as follows: A 1 µL aliquot of as-isolated protein (10 mg/mL in 100 mM Tris-HCl pH 8) was combined with 1 µL of precipitant solution (150–250 mM $MgCl_2$, 16–20% (w/v) PEG 3350) on a glass cover slide and sealed over a reservoir containing 500 µL of precipitant solution. Diffraction quality crystals grew in 1–6 d. Crystals were soaked in a cryo-protectant solution containing 250 mM $MgCl_2$, 18–20% (w/v) PEG 3350, 9% (v/v) glycerol and cryo-cooled in liquid nitrogen.

To reduce crystals of as-isolated DvCODH, crystals were transferred into a soaking solution containing 250 mM $MgCl_2$, 18% (w/v) PEG 3350, 5 mM sodium dithionite and incubated for 30 min. For structures of reduced DvCODH, crystals were transferred to a cryo-protectant solution containing 250 mM $MgCl_2$, 18% (w/v) PEG 3350, 9% (v/v) glycerol and cryo-cooled in liquid nitrogen. For structures of reduced and then oxygen-exposed DvCODH, crystals were transferred into a dithionite-free drop containing 250 mM $MgCl_2$, 18% (w/v) PEG 3350 prior to removal from the anaerobic chamber to avoid reaction of excess dithionite with molecular oxygen. Following removal from the chamber, 0.5 µL of aerobically-prepared precipitant solution was added to the drop to initiate equilibration with ambient atmospheric conditions. Crystals were harvested after 2 d as described above.

## Data collection, model building, and refinement

All data were collected at the Advanced Photon Source (Argonne, IL) at beamline 24-ID-C at a temperature of 100 K using a Pilatus 6M pixel detector. Where applicable, native, Fe peak, and Ni peak data were collected on the same crystal for a particular sample. Native data were collected at an energy of 12662 eV (0.9792 Å); Fe peak data at 7130 eV (1.7389 Å); and Ni peak data at 8360 eV (1.4831 Å). All data were integrated in XDS and scaled in XSCALE (*Kabsch, 2010*). Data collection statistics are summarized in *Supplementary file 1*.

The initial structure of DvCODH was determined to 2.50 Å resolution by molecular replacement (MR) in the program Phaser (*McCoy et al., 2007*) using data from crystals belonging to space group $P2_12_12_1$. The search model for MR was generated from the structure of the CODH from *Rhodospirillum rubrum* (47% sequence identity; PDB ID: 1JQK) by modification in Sculptor (*Bunkóczi and Read, 2011*) using the Schwarzenbacher algorithm (*Schwarzenbacher et al., 2004*) with a pruning level of 2 to truncate non-identical residues at the Cβ position. Metalloclusters were not included in the search model. A single MR solution was found with an LLG of 311, TFZ of 21.3, and R-value of 57.9. The model was completed through iterative rounds of model building in Coot (*Emsley et al., 2010*) and refinement in Phenix (*Adams et al., 2010*) (see below). Subsequent structures were determined by MR in Phaser using the initial DvCODH structure as a search model. Following MR, 10 cycles of simulated annealing refinement were performed in Phenix to eliminate existing model bias.

For all structures, refinement of atomic coordinates and atomic displacement parameters (B-factors) was carried out in Phenix using noncrystallographic symmetry (NCS) restraints. Models were completed by iterative rounds of model building in Coot and refinement in Phenix. In advanced stages of refinement, water molecules were added automatically in Phenix and modified in Coot with placement of additional water molecules until their number was stable. Final stages of refinement included translation, libration, screw (TLS) parameterization with one TLS group per monomer (*Painter and Merritt, 2006*). For structures determined to less than or equal to 2 Å resolution, NCS restraints were removed in final refinement cycles.

In advanced stages of structural refinement of the 1.72 Å as-isolated DvCODH structure, it became clear that two conformations of the C-cluster were present. Based on the electron density, the Fe-S scaffold of the oxidized form of the cluster was modeled at an occupancy of 80%, with the

Ni ion at 70%. The canonical, reduced form of the cluster without Ni was modeled with an atomic occupancy of 20%. A peak of residual electron density at a position bridging $Fe_u$ and $Fe_1$ of the oxidized cluster appeared in late stages of refinement. Modeling of a water molecule at this position resulted in a refined occupancy of ~30%. The geometry of this site, however, is not consistent with coordination of $H_2O/OH^-$, and given the long Fe-ligand bond distances (2.4 Å), the site is likely occupied by a heavier atom, for example $Cl^-$ from the protein buffer. Due to the low occupancy (<30%) of an atom heavier than water at this site and the inability to resolve the identity of this ligand crystallographically, this site was left unmodeled in the final structure.

The structure of $Dv$CODH(C301S) also contained an apparent mixture of cluster types at the C-cluster site. Here, the [3Fe-4S] partial cubane portion of the canonical C-cluster is intact and present at full occupancy (*Figure 5—figure supplement 1*); however, modeling of $Fe_u$ proved complicated. When modeled and refined as a [3Fe-4S]-$Fe_u$ cluster at full occupancy, the atomic displacement parameter (*B*-factor) of $Fe_u$ was higher (38.7 Å$^2$) than the average for the ligating atoms of His266 (Nε) and Cys302 (Sγ) (22.9 Å$^2$) as well as for the remainder of the cluster (24.0 Å$^2$), suggesting that $Fe_u$ may be present at reduced occupancy. Additionally, positive difference electron density ($F_o-F_c$) near the Ni-binding site of the C-cluster was observed, indicating the presence of an atom in this site (*Figure 5—figure supplement 1a*). Based on the ICP-OES results, $Dv$CODH(C301S) does not contain Ni, suggesting that an atom other than Ni occupies this site in the structure. Anomalous difference maps calculated from diffraction data collected at the iron peak wavelength (7130 eV) revealed a shoulder extending from $Fe_u$ into this site, indicative of the presence of Fe at partial occupancy within the cubane (*Figure 5—figure supplement 1b*). Together, the native diffraction data, anomalous difference data, and *B*-factor analysis suggested that there are two different states of the C-cluster in the sample: one with the canonical [3Fe-4S]-$Fe_u$ scaffold and one in the form of a distorted [4Fe-4S] cubane. Indeed, when $Fe_u$ is modeled with a split conformation such that at 70% occupancy it is present in its unique binding site and at 30% occupancy it is incorporated into the cubane, the cluster refines well into the electron density and the *B*-factors of $Fe_u$ are better matched with those of the surrounding atoms (*Figure 5*).

Final refinement of each structure yielded models with low free *R*-factors, excellent stereochemistry, and small root mean square deviations from ideal values for bond lengths and angles. All refinement statistics are summarized in *Supplementary file 1*. Side chains without visible electron density were truncated to the last atom with electron density and amino acids without visible electron density were not included in the models. Final models contain the following residues (of 629 total): as-isolated (batch 1): 4–628 (chains A and B); as-isolated (batch 2): 4–629 (chain A), 2–629 (chain B); reduced (batch 2): 4–627 (chain A), 4–628 (chain B); reduced/$O_2$-exposed (batch 2): 8–63, 68–286, 289–629 (chain A), 6–63, 67–287, 291–628 (chain B); $Dv$CODH(C301S): 4–628 (chain A), 4–628 (chain B), 5–627 (chain C), 3–628 (chain D). Models were validated using simulated annealing composite omit maps calculated in Phenix. Model geometry was analyzed using MolProbity (*Chen et al., 2010*). Analysis of Ramachandran statistics indicated that each structure contained the following percentages of residues in the favored, allowed, and disallowed regions, respectively: as-isolated (batch 1): 96.3%, 3.4%, 0.3%; as-isolated (batch 2): 96.8%, 2.9%, 0.3%; reduced (batch 2): 96.6%, 3.1%, 0.3%; reduced/$O_2$-exposed (batch 2): 96.5%, 3.2%, 0.3%; $Dv$CODH(C301S): 97.0%, 2.7%, 0.3%. Figures were generated in PyMOL (*Schrodinger, 2015*). Crystallography packages were compiled by SBGrid (*Morin et al., 2013*).

## EPR spectroscopy sample preparation and data collection

EPR samples were prepared using $^{57}$Fe-enriched $Dv$CODH containing 13.3 $^{57}$Fe/monomer and 0.5 Ni/monomer, as quantified by ICP-OES. All samples were prepared under oxygen-free conditions in a Coy anaerobic chamber. Samples were incubated with an excess of sodium dithionite (30–40 equivalents) for 20–30 min at 22 °C prior to freezing in liquid $N_2$ under oxygen-free conditions. For the sample of air-exposed and re-reduced $Dv$CODH, an aliquot was removed from the anaerobic chamber and incubated on ice under ambient atmospheric conditions for 50 min to afford full oxidation of the clusters. The sample was then returned to the anaerobic chamber and incubated with 30–40 equivalents of sodium dithionite for 20–30 min. Samples (250 μL) were loaded in Quartz EPR tubes (QSI Inc, Fairport Harbor, OH) and frozen in liquid $N_2$ under oxygen-free conditions. EPR spectra were acquired at the Department of Chemistry Instrumentation Facility at MIT on a Bruker EMX Plus continuous wave (CW) X-Band spectrometer (operating at ~9.34 GHz) equipped with a

rectangular resonator ($TE_{101}$) and a cryogen-free system consisting of a Sumitomo RDK-408D2 cold head equipped with a ColdEdge Technologies waveguide cryostat. Spectra were acquired using Bruker Xenon software and were recorded at 10 K at a microwave power of 0.2 mW, using a modulation amplitude of 1 mT, a microwave frequency of 9.34 GHz, a conversion time of 82.07 ms, and a time constant of 81.92 ms. Spin quantification was carried out against a $Cu^{2+}$-EDTA standard containing 200 µM $CuSO_4$ in 10 mM EDTA, under non-saturating conditions. Quantitation of the $S = 1/2$ [Fe-S] centers amounted to 3.2 spins/dimer, similar to our previous report on $Dv$CODH (*Hadj-Saïd et al., 2015*). Quantification of the $C_{red1}$ and $C_{red2}$ states was carried out on the basis of numerical double-integration of the simulated spectra using the MATLAB-based EasySpin software (*Stoll and Schweiger, 2006*); $C_{red1}$ was 0.43 spins/dimer and $C_{red2}$ was 0.35 spins/dimer.

## Acknowledgements

The authors thank Steven Cohen (MIT) for assistance with EPR data collection. We additionally thank David Born and Tsehai Grell (both MIT) for helpful conversations. This work was supported by National Institutes of Health (NIH) grants T32 GM008334 (ECW), R01 GM069857 and R35 GM126982 (CLD), and R00 GM111978 (M-EP); and ANR projects MeCO2Bio and SHIELDS. CLD is a Howard Hughes Medical Institute Investigator and a senior fellow of the Bio-inspired Solar Energy Program, Canadian Institute for Advanced Research. This work is based on research conducted at the Advanced Photon Source on the Northeastern Collaborative Access Team beamlines, which are funded by the National Institute of General Medical Sciences from the NIH (P41 GM103403). The Pilatus 6M detector on beamline 24-ID-C is funded by a NIH Office of Research Infrastructure Programs High End Instrumentation grant (S10 RR029205). This research used resources of the Advanced Photon Source, a U.S. Department of Energy (DOE) Office of Science User Facility operated for the DOE Office of Science by Argonne National Laboratory under Contract No. DE-AC02-06CH11357. EPR data were collected at the MIT Department of Chemistry Instrumentation Facility.

## Additional information

### Funding

| Funder | Grant reference number | Author |
| --- | --- | --- |
| National Institutes of Health | R01 GM069857 | Catherine L Drennan |
| Howard Hughes Medical Institute | | Catherine L Drennan |
| Canadian Institute for Advanced Research | | Catherine L Drennan |
| National Institutes of Health | T32 GM008334 | Elizabeth C. Wittenborn |
| National Institutes of Health | R00 GM111978 | Maria-Eirini Pandelia |
| Agence Nationale de la Recherche | ANR-17-CE11-0027 | Christophe Léger Vincent Fourmond Sébastien Dementin |
| National Institutes of Health | R35 GM126982 | Catherine L Drennan |

The funders had no role in study design, data collection and interpretation, or the decision to submit the work for publication.

### Author contributions

Elizabeth C Wittenborn, Formal analysis, Validation, Investigation, Visualization, Methodology, Writing—original draft, Project administration, Writing—review and editing, Performed the crystallographic experiments, Analyzed the crystallographic data, Wrote the manuscript; Mériem Merrouch, Laura Fradale, Formal analysis, Validation, Investigation, Purified protein and performed activity assays; Chie Ueda, Formal analysis, Validation, Investigation, Visualization, Prepared EPR samples, Analyzed the EPR data; Christophe Léger, Conceptualization, Resources, Supervision, Funding acquisition, Methodology, Project administration, Writing—review and editing, Assisted with editing

the manuscript; Vincent Fourmond, Conceptualization, Supervision, Funding acquisition, Methodology, Project administration, Writing—review and editing, Assisted with editing the manuscript; Maria-Eirini Pandelia, Resources, Formal analysis, Supervision, Funding acquisition, Validation, Investigation, Visualization, Methodology, Project administration, Writing—review and editing, Collected and analyzed the EPR data, Assisted in preparing the manuscript; Sébastien Dementin, Conceptualization, Resources, Formal analysis, Supervision, Funding acquisition, Validation, Investigation, Visualization, Methodology, Project administration, Writing—review and editing, Purified protein, Performed activity assays, Assisted in editing the manuscript; Catherine L Drennan, Conceptualization, Resources, Supervision, Funding acquisition, Methodology, Writing—original draft, Project administration, Writing—review and editing, Helped analyze the crystallographic data, Wrote the manuscript

### Author ORCIDs
Elizabeth C Wittenborn http://orcid.org/0000-0002-8473-0814
Christophe Léger https://orcid.org/0000-0002-8871-6059
Vincent Fourmond http://orcid.org/0000-0001-9837-6214
Maria-Eirini Pandelia https://orcid.org/0000-0002-6750-1948
Catherine L Drennan http://orcid.org/0000-0001-5486-2755

### Decision letter and Author response
Decision letter https://doi.org/10.7554/eLife.39451.030
Author response https://doi.org/10.7554/eLife.39451.031

## Additional files

### Supplementary files
• Supplementary file 1. Crystallographic data collection and refinement statistics.
DOI: https://doi.org/10.7554/eLife.39451.016
• Supplementary file 2. Metal content and activity of DvCODH preparations.
DOI: https://doi.org/10.7554/eLife.39451.017
• Transparent reporting form
DOI: https://doi.org/10.7554/eLife.39451.018

### Data availability
Crystallographic model coordinates and structure factors have been deposited in the Protein Data Bank (www.rcsb.org) under the following accession codes: 6B6V (as-isolated DvCODH, batch 1), 6B6W (as-isolated DvCODH, batch 2), 6B6X (reduced DvCODH, batch 2), 6B6Y (reduced/O2-exposed DvCODH, batch 2), and 6DC2 (DvCODH(C301S)).

The following datasets were generated:

| Author(s) | Year | Dataset title | Dataset URL | Database, license, and accessibility information |
|---|---|---|---|---|
| Elizabeth C Wittenborn, Catherine L Drennan | 2017 | Crystal structure of Desulfovibrio vulgaris carbon monoxide dehydrogenase, as-isolated (protein batch 1), canonical C-cluster | http://www.rcsb.org/structure/6B6V | Publicly available at the RCSB Protein Data Bank (accession no: 6B6V) |
| Elizabeth C Wittenborn, Catherine L Drennan | 2017 | Crystal structure of Desulfovibrio vulgaris carbon monoxide dehydrogenase, as-isolated (protein batch 2), oxidized C-cluster | http://www.rcsb.org/structure/6B6W | Publicly available at the RCSB Protein Data Bank (accession no: 6B6W) |
| Elizabeth C Wittenborn, Catherine L Drennan | 2017 | Crystal structure of Desulfovibrio vulgaris carbon monoxide dehydrogenase, dithionite-reduced (protein batch 2), canonical C-cluster | http://www.rcsb.org/structure/6B6X | Publicly available at the RCSB Protein Data Bank (accession no: 6B6X) |
| Elizabeth C Witten- | 2017 | Crystal structure of Desulfovibrio | http://www.rcsb.org/ | Publicly available at |

| | | | | |
|---|---|---|---|---|
| born, Catherine L Drennan | | vulgaris carbon monoxide dehydrogenase, dithionite-reduced then oxygen-exposed (protein batch 2), oxidized C-cluster | structure/6B6Y | the RCSB Protein Data Bank (accession no: 6B6Y) |
| Elizabeth C Wittenborn, Catherine L Drennan | 2018 | Crystal structure of Desulfovibrio vulgaris carbon monoxide dehydrogenase C301S variant | http://www.rcsb.org/structure/6DC2 | Publicly available at the RCSB Protein Data Bank (accession no: 6DC2) |

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
