## [Decision Letter]

Thank you for submitting your article "Redox-dependent metal migration in the Ni-Fe-S cluster of carbon monoxide dehydrogenase" for consideration by *eLife*. Your article has been reviewed by three peer reviewers, including Jon Clardy as the Reviewing Editor and Reviewer #1, and the evaluation has been overseen by Philip Cole as the Senior Editor. The following individual involved in review of your submission has agreed to reveal their identity: Douglas C Rees (Reviewer #3).

The reviewers have discussed the reviews with one another and the Reviewing Editor has drafted this decision to help you prepare a revised submission.

All of the reviewers were strongly in favor of publication. They were uniformly impressed with the importance of the findings and the technical rigor with which the studies were performed. The major issues that they felt needed to be addressed were with the presentation, which was more suitable for a specialized audience rather than the general readership that *eLife* seeks to attract. The specific issues were:

1) Most readers are highly unlikely to be familiar with any previous work on carbon monoxide dehydrogenase, Fe-S enzymes, Ni-Fe-S-enzymes, bioenergy, or 'great clusters'. The manuscript as currently written dives right in, and it would be highly desirable to begin a bit further back. They suggested beginning with a discussion of Fe-S enzymes and electron transfer in the Introduction – something to take a reader who knows the rudiments of enzymology into the exciting world of electron transfer reactions. We would also suggest changing the title to "Redox-dependent rearrangements of the NIFeS cluster in CO dehydrogenase" or something similar.

2) In line with the previous suggestion all reviewers felt that the Figure 1 supplement should be moved into the article proper. It summarizes the findings of the paper in terms that are likely to be more familiar to the general reader. Also it would be helpful to have a more explicit depiction of the redox states of C_ox_ and C_red_ in the figure and text.

---

## [Author Response]

All of the reviewers were strongly in favor of publication. They were uniformly impressed with the importance of the findings and the technical rigor with which the studies were performed. The major issues that they felt needed to be addressed were with the presentation, which was more suitable for a specialized audience rather than the general readership that eLife seeks to attract. The specific issues were:1) Most readers are highly unlikely to be familiar with any previous work on carbon monoxide dehydrogenase, Fe-S enzymes, Ni-Fe-S-enzymes, bioenergy, or 'great clusters'. The manuscript as currently written dives right in, and it would be highly desirable to begin a bit further back. They suggested beginning with a discussion of Fe-S enzymes and electron transfer in the Introduction – something to take a reader who knows the rudiments of enzymology into the exciting world of electron transfer reactions. We would also suggest changing the title to "Redox-dependent rearrangements of the NIFeS cluster in CO dehydrogenase" or something similar.

We agree that our previous version of the manuscript did not give enough background information for a more general audience. We have rewritten the Introduction to the paper to highlight key aspects of Fe-S cluster enzymes, the “great clusters”, and carbon monoxide dehydrogenase. We feel that this new Introduction enhances the quality of our manuscript and thank the reviewers for their insight. We have also changed the title of the manuscript in line with the reviewers’ suggestion.

2) In line with the previous suggestion all reviewers felt that the Figure 1 supplement should be moved into the article proper. It summarizes the findings of the paper in terms that are likely to be more familiar to the general reader. Also it would be helpful to have a more explicit depiction of the redox states of C_ox_ and C_red_ in the figure and text.

We have moved the indicated figure into the main text as Figure 2. This figure now includes a depiction of the C_ox_ state, and a clearer explanation of the various C-cluster redox states has been included in our revised Introduction.